# A Case study on optimal Deep Learning model for UAVs

## Abstract

Over the passage of time Unmanned Autonomous Vehicles (UAVs), especially Autonomous flying drones grabbed a lot of attention in Artificial Intelligence. Since electronic technology is getting smaller, cheaper and more efficient, huge advancement in the study of UAVs has been observed recently. From monitoring floods, discerning the spread of algae in water bodies to detecting forest trail, their application is far and wide. Our work is mainly focused on autonomous flying drones where we establish a case study towards efficiency, robustness and accuracy of UAVs where we showed our results well supported through experiments. We provide details of the software and hardware architecture used in the study. We further discuss about our implementation algorithms and present experiments that provide a comparison between three different state-of-the-art algorithms namely TrailNet, InceptionResnet and MobileNet in terms of accuracy, robustness, power consumption and inference time. In our study, we have shown that MobileNet has produced better results with very less computational requirement and power consumption. We have also reported the challenges we have faced during our work as well as a brief discussion on our future work to improve safety features and performance.

## 1 Introduction

In modern era, UAVs have become very popular and have basic intelligence of being driven autonomously. Talking about ground traffic, these vehicles have limitations of physical paths and barriers. However, such is not the case with flying objects like drones as they do not suffer from such physical limitations. Autonomous Flying objects are in much of discussion these days and are striding all across the realm - traffic monitoring Kanistras et al. (2013), agriculture Reinecke & Prinsloo (2017), inventory management Bae et al. (2016), surveillance Zaheer et al. (2016), data mining, disaster response Luo et al. (2015), etc. As their areas of application increase, it becomes more important to find algorithms well suited for these kind of vehicles. Some applications may not require the drone to be extremely accurate, but may require it to work for longer durations e.g. in surveillance applications while others may require it to be very precise but may not require it to work for long duration e.g. in delivery of items.

In the last decade, significant changes have been observed in the field of autonomous motion planning of vehicles, UAVs in particular. The motion planning of UAVs are distinctly difficult because of several complexities that comes with aerial vehicles. The salience of differential constraints, uncertainty in the vehicle state and limited knowledge about the environment makes it impossible to have a precise pre-computed plan to follow through. These differences considerably gave rise to various approaches and techniques for planning the motion of these unmanned autonomous vehicles. The ambiguity of different algorithms and its inherent adversity pose an intriguing scope for a bench-marking study to enhance accuracy, robustness, power consumption, safety and inference time and fine tuning it further.

Throughout this paper, we bear in mind some of the generic characteristics and prerequisites relating to UAVs. The basic design of a UAV is modelled to have acceleration and velocity constraints. Furthermore, the higher-order differential constraints also associate themselves with the equation of motion of a drone. However, the coherent objective involved in all UAVs is to guide the vehicle

towards a goal. In this paper, we introduce to the best of our knowledge, a very first comparative study of three algorithms in order to find a better motion control of a drone for detecting a trail.

In order to be able to compare a set of algorithms in a meticulous way, it is necessary to establish their precision and robustness and to evaluate its power consumption as well as inference time. Along with these metrics established for a particular algorithm, it is also necessary to consider the distinct areas of its application. Only then, based on the requirements called for by a particular application, a reasonable opinion about an algorithm is to be formed. Our study covers recent developments and algorithms used in the area of trail detection by UAVs and runs down as comprehensively as possible what has been already upheld regarding these algorithms.

## 2 RELATED WORK

We briefly review the recent applications of drones and Unmanned Aerial Vehicles(UAVs) and their challenges. Our work is partly inspired by the research of Smolyanskiy et al. (2017) on a MAV(Micro Aerial Vehicles) system for autonomous trail. They introduced and trained a deep neural network named Trailnet to estimate the view orientation and lateral offset of the MAV with respect to the trail center. They ran all their vision systems in real time on NVIDIA Jetson TX1. We have presented a detailed study on how we can increase their accuracy and lower their training time. We selected InceptionResnet Szegedy et al. (2016) and MobileNet Howard et al. (2017) over their proposed Trailnet Smolyanskiy et al. (2017).

Inception architecture Szegedy et al. (2016) has been shown to achieve very good performance at relatively low computational cost. Recently, the introduction of residual connections in conjunction with a more traditional architecture has yielded state-of-art performance in the 2015 ILSVRC challenge. Its performance was similar to the latest generation Inception-v3 network. There is also some evidence of residual Inception networks outperforming similarly expensive Inception networks without residual connections by a thin margin.

In Howard et al. (2017) a class of efficient models called MobileNets was first presented for embedded vision applications. They are based on a streamlined architecture which uses depth-wise separable convolutions to build light weight deep neural networks. Two simple global hyper-parameters introduced efficiently trade off between latency and accuracy. MobileNets are built primarily from depth-wise separable convolutions initially introduced in Sif and subsequently used in Inception models (Ioffe & Szegedy (2015)) to reduce the computation in the first few layers.

## 3 METHODOLOGY

In this paper, we have tried to put forward a comparative study to find an algorithm well suited for UAVs. For this task, we selected 3 algorithms from state-of-the-art namely Trailnet, Inception-Resnet and Mobilenet. We have chosen Inception-ResNet and MobileNet for our experiments as they were proven to perform very well on classification tasks. Our aim is to train these algorithms offline, implement them on the drone and use their output commands to correctly navigate the drone through trails. We have used Udacity simulator to simulate the path generated by the output of these algorithms and to compare their robustness during long trails.

### 3.1 SYSTEM ARCHITECTURE

An important goal of our method is to demonstrate the effectiveness of low cost systems for the complex task of flying an autonomous drone. In this section, we want to put forward the architecture (both hardware and software) that we have used for our research in order to achieve our result. We will further discuss about the basis of choosing Jetson TX2 for computing and how it has shaped our upshot. Also, we will talk about using ROS together with Ubuntu L4T and the hardware configuration of our drone.

#### 3.1.1 HARDWARE ARCHITECTURE

1. Drone Setup: Our drone comprises of DJI Flame Wheel F450 quadcopter with an open source Pixhawk autopilot mounted on a Orbitty carrier board.

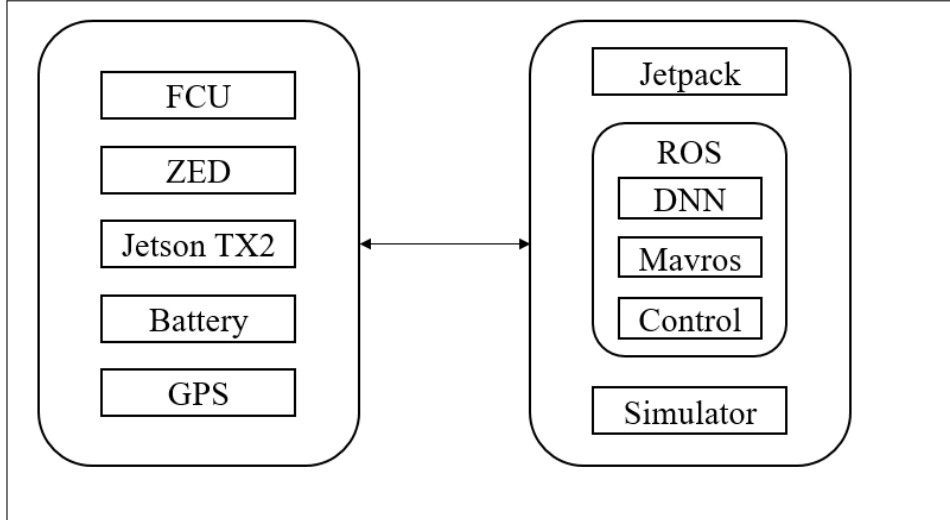

Figure 1: System Architecture

2. Jetson TX2: We use NVIDIA Jetson TX2 nvi, the fastest supercomputing edge computing device for interfacing. Jetson TX2 is NVIDIAs second-generation CUDA-capable edge device nvi. Like its predecessor TX1 nvi, TX2 runs on Ubuntu Linux4Tegra(L4T) and JetPack-L4T 3.2. It features an integrated 256 CUDA core NVIDIA Pascal GPU, a hex-core ARMv8 65-bit CPU complex, and 8GB of LPDDR4 memory with a 128-bit interface. The CPU complex shown in Fig blends a quad-core ARM Cortex-A57 with a dual-core NVIDIA Denver 2 (nvi (2017)).We have also used a Windows 10, Intel(R) Xeon(R) Silver 4114 CPU @2.20GHz(20 CPUs), 2.2GHz, 32GB RAM with Nvidia Titan Xp GPU for training purposes.

3. ZED Camera: We have used ZED stereo camera, which is a standard 3D USB camera for object detection and image capturing. It adds depth perception, positional tracking and 3D mapping using advanced sensing technology based on human stereo vision.

4. Battery: We use Venom 35C, 14.8V, 5000 mAh lithium polymer battery which is best suitable for our drone

5. GPS: We use Holybro Ublox Neo-M8N GPS module which provides high sensitivity and minimal acquisition time while maintaining low system power.

### 3.1.2 SOFTWARE ARCHITECTURE

1. Jetpack - Jetpack is an SDK released by NVIDIA and is the most comprehensive solution for building artificially intelligent applications in recent times jet (2014). Our Jetson Developer Kit (TX2) on the drone was first flashed with the latest OS image - Ubuntu 16.04 (Xenial). We further used JetPack installer to install developer tools for both Host and the Drone (the developer kit on the drone). All the libraries, TensorRT, cuDNN, CUDA, OpenCV to name some were installed to jumpstart our development environment. For our case study, we used JetPack-L4T 3.2.

2. ROS - ROS or Robot Operating System is a collection of Linux based frameworks acting as robotics middleware. We have used ROS Kinetic Kame kin for hardware abstraction of Joystick and the Controller present on the drone. ROS also plays an important part in controlling devices at a very low-level as well as for transferring data. A ROS system typically consists of a number of independent nodes. For our environment, these nodes are MAVROS, Control Node, DNN, Camera and Joystick - each of which is able to communicate with each other using subscribe or publish messaging model. All nodes are registered with the master node (MAVROS for our case) which in turn helps them to find and communicate with each other. The MAVROS enables MAVLink (Micro Air Vehicle Link) mav (2018) protocol to communicate with the PX4 (FCU) on-board.

3. Communication - The on-board Jetson TX2 had wifi access point enabled by us before installing it on the drone. As a result, the host PC could connect to the Jetson TX2 wirelessly and was able to access it remotely. By sending commands through the terminal, the host was able to control the drone by Secure Socket Shell (SSH) network protocol.

4. Udacity Simulator - Learning about UAVs comes with a lot of risks considering the constant fright of crashing. From testing new hardwares, learning to control the flight controller unit of the drone to further fine tuning of our algorithms, the resulting failures can be really expensive. To overcome this we have used a simulator offered by Udacity in its Flying Car nanodegree program. Figure 2 demonstrates the state transition diagram of the drone using this simulator.

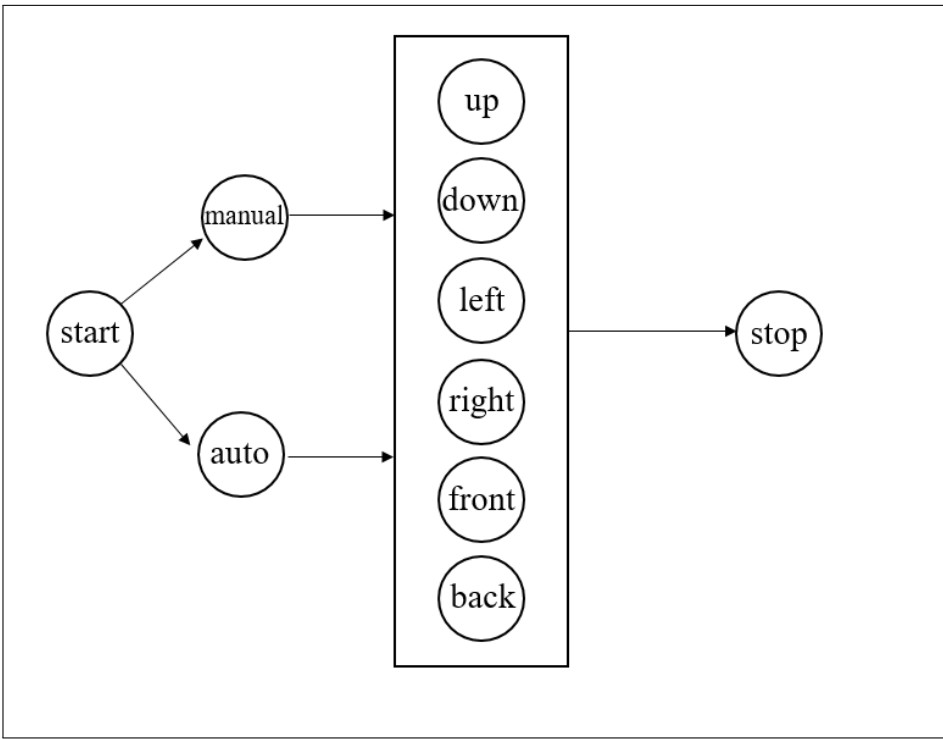

Figure 2: State transition diagram of the drone in udacity simulator

## 3.2 CONTROLS

We use standard UART(Universal Asynchronous Receiver/Transmitter) communications for controls. The transmitting UART converts parallel data from a controlling device like CPU into serial form, transmits it in serial form to receiving UART, which then converts the serial data back to parallel for the receiving device. Figure 3 shows the workflow involving all the modules and communication between them.

## 3.3 ALGORITHM

For our purposes, we have used 3 different algorithms namely MobileNet Howard et al. (2017) , Inception-Resnet Szegedy et al. (2016) and Trailnet Smolyanskiy et al. (2017). The choice of algorithm is dependent on performance of state-of-the-art algorithms on Imagenet Deng et al. (2009).

## 3.4 DATASETS

For our experiments, we use IDSIA trail dataset Giusti et al. (2016) which were collected on Swiss Alps forest trails. It consists of approximately 15GB of image data acquired using different cameras.

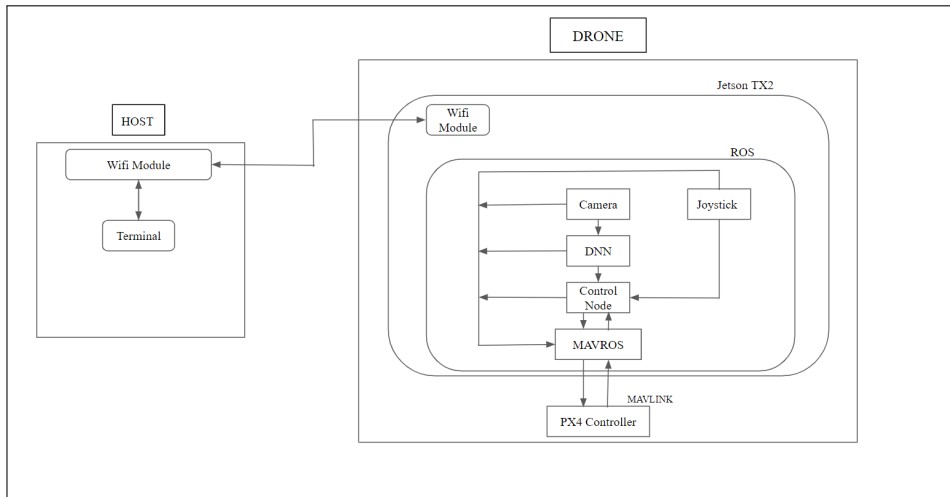

Figure 3: Workflow and communication between the host PC and drone.

The entire dataset is divided into 15 parts, in the form of folders numbered from 000 till 014. We have used folders 001, 002, 004, 005, 006, 007 and 009 for training our model and used folders 003, 008 and 010 for validation. We ran our tests on folder 012. We had to discard folders 000, 013 and 014 as they comprise of only preliminary test data.

# 4 EXPERIMENTS

In this section we first go through the model selections that we made by considering their performances on ImageNet Challenge Russakovsky et al. (2015). ImageNet project in an ongoing large visual database designed for use in visual object recognition software research. Since 2010, ImageNet has been running an annual competition in visual recognition where participants are provided with 1.2 Million images belonging to 1000 different classes. Several deep learning architectures have been proposed since then and we have considered two of them for our experiments.

## 4.1 MODEL CHOICES

We have chosen the models based on their performance on Imagenet Challenge Russakovsky et al. (2015) considering accuracy and computation as major metrics. Accuracy is one of the critical aspect in building deep learning models. It depends on network architecture and the amount of data available for training. Also most of the convolutional neural networks (ConvNets) have huge memory and computational requirements especially during training. This is an important concern specifically for our purpose as memory and space footprint is limited in embedded AI devices such as drones. Also size of the final trained model becomes important to consider as we will be deploying the models to run locally on drone. In general, more computationally intensive networks usually tends to produce more accuracy. Hence, there is always a trade-off between accuracy and computational cost.

Apart from these, there are many other factors which are important in selecting models such as training time, ability of a network to generalize well, inference time etc. Considering all of these factors, we have chosen MobileNet Howard et al. (2017) and Inception-Resnet pretrained Szegedy et al. (2016) on ImageNet Deng et al. (2009) for our experiments. Table 1 compares these model factors on ImageNet project.

## 4.2 TRANSFER LEARNING AND FINE-TUNING

Transfer learning is a Machine Learning technique to store knowledge gained on solving one problem and applying it to a different related problem. The three major transfer learning scenarios are

Table 1: Performance of Inception-ResNet and MobileNet on ImageNet.

| DNN | Parameters | Top 1 Accuracy (in %) | Top 5 Accuracy (in %) |
|---|---|---|---|
| Inception-Resnet | 54M | 80.4 | 95.3 |
| Mobilenet | 3.5M | 70.1 | 89.9 |

Table 2: A comparison of 3 algorithms on different parameters.

| DNN | No. of Layers | No. of Parameters(in Millions) | Training Time | Accuracy(in %) |
|---|---|---|---|---|
| Trailnet | 18 | 10M | 13 hours [1] | 84 |
| Inception-Resnet | 27 | 54M | 8 hours | 93 |
| Mobilenet | 28 | 3.5M | 2 hours | 89 |

using ConvNet as fixed feature extractor, fine-tuning the ConvNet and using pre-trained models. The two most important factors that help us to decide on what type of transfer learning we should perform on new dataset are size of the dataset and similarity to original dataset. Since our dataset is large enough and different from ImageNet Deng et al. (2009) dataset, we have decided to fine-tune our models.

Fine-tuning is a transfer learning strategy to replace the final layers of the ConvNets and tweak them so that they can learn more robust features relevant to our problem. We then retrain the classifier on the new dataset and fine-tune the weights of the pre-trained network by continuing the back propagation. Usually the earlier features of a ConvNet contain more generic features ( like edge detectors, color detectors etc. ) which are useful for many image detecting tasks. Later layers of ConvNets extract minute detailed features specific to the classes contained in our problem.We have fine-tuned both InceptionResnet and MobileNet by initializing them with their weights pretrained on ImageNet Deng et al. (2009) and retrain the model from scratch.

### 4.3 RESULTS

In this section we discuss the results of our experiments with comparisons in terms of size of the models, accuracy, inference time and power consumption.

#### 4.3.1 SIZE AND ACCURACY

Table 2 compares the architectural complexity, training time and accuracy of TrailNet, Inception-Resnet and MobileNet on our dataset. It is evident from our observations that both InceptionResnet and MobileNet have an accuracy better than TrailNet. Also the training time and computational cost involved for MobileNet is much less than that of other two models because of its less complexity, even though its accuracy is on par with InceptionResnet model.

We have performed testing of our models by running them on Udacity simulator. We have used a dataset consisting of 2000 images pertaining to a trail path of approximately 100 meters and simulated the drone by the commands given by output of the algorithms. Figure 4 shows the path traversed by the drone autonomously on the test data environment, where a) is the path traversed by drone manually controlled using simulator, b) is the path traversed when the simulator is controlled with output of Inception-Resnet and c) is the path traversed by drone when controlled with output of MobileNet. It can be seen that the drone is more robust and pretty much followed the ground truth path using both the models.

#### 4.3.2 INFERENCE TIME & POWER CONSUMPTION

We have measured the inference time (i.e., time taken by the model to predict output of test dataset) on Jetson TX2 machine and the following table shows the comparison. From our observations in Table 3, MobileNet has a very less inference time, which means the cameras with high frame rate can

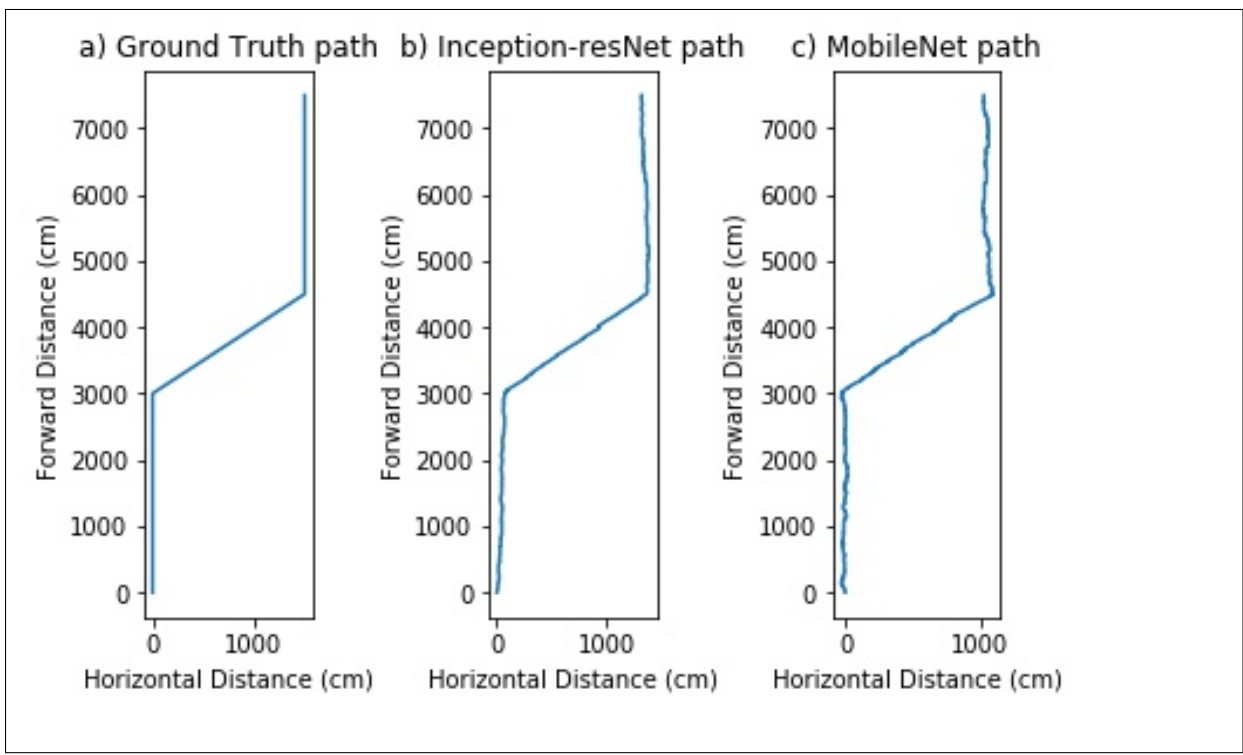

Figure 4: Graph depicting path traversed by the drone simulator under different algorithms.

Table 3: A comparison of 3 algorithms on inference time and power.

| DNN | Inference time (in ms) | Peak Power consumed (Training, in W) | Inference power draw (in W) |
|---|---|---|---|
| Trailnet | 49.6 | N/A* | 110 |
| Inception-Resnet | 55.15 | 260 | 120 |
| Mobilenet | **21.73** | **210** | **65** |

be operated using the MobileNet model for more time and with less energy consumption compared to other two models.

We have calculated our power draw values on Nvidia Titan Xp GPU with a maximum possible power draw of 300W. Also the average power drawn with idle GPU is 15W. We can see that MobileNet draws very less power compared to other two models which results in long battery life of the drone. We have taken a batch size of 32 for inference power draw calculation.

From our observations in Table 2, MobileNet has a very less inference time, which means the cameras with high frame rate can be operated using the MobileNet model for more time and with less energy consumption compared to other two models. Note that we have not trained TrailNet model from scratch and so we have not reported the peak power consumed in this case.

## 5 CHALLENGES & DISCUSSION

While producing this work, we have encountered several challenges. Few of these challenges are listed below:

1. The major challenge encountered was to run our DNN models on the physical drone in real time due to a hardware bug we were facing with the FCU.

2. We could not train our models from scratch due to lack of significant amount of dataset. Additionally, we handled a lot of issues to make our models more stable and robust. Since in each trial, the number of images in each class (left, straight and right) were different, there was a lot of data imbalance which we solved by upsampling and downsampling the dataset.

3. Due to the large number of training parameters at the beginning, our models were over-fitted. We eliminated over-fitting by introducing several data augmentation techniques (random flipping, random rotation, random contrast and transition etc. ). We further included regularization (especially dropout layers) in order to reduce network complexity.

4. Power is one of the important factors especially in mobile embedded devices with small size and computational power. Typically, deep learning algorithms consume more power specifically for the real time inference. We have made an estimate of the power consumption of each of our model by calculating the GPU power drawn by them but we could not test how long our drone would run implementing each of these models due to the hardware bug mentioned before.

## 6 CONCLUSION & FUTURE WORK

In this paper, we have presented a comparison between 3 algorithms - TrailNet, InceptionResnet and MobileNet in terms of accuracy, computational cost, power consumption, inference time and robustness. The choice of algorithm for UAVs varies on the basis of several factors. In our work, we have worked with some of the factors which we thought would be pivotal in algorithm selection considering reasonable comparisons. We observed in our study that MobileNet outperformed others with very less computational requirement and power consumption. Hence in our opinion, MobileNet is more befitting for drones and other embedded devices compared to TrailNet and InceptionResnet.

Safety is another major concern in terms of drones. There can be many situations such as collision with the objects, external disturbances like winds, chances of drone moving out of manual controller zone, battery issues, chances of getting stolen and other safety hazards. We will be implementing these drone related safety features in our future work.

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
