# OpenReview forum: "A CASE STUDY ON OPTIMAL DEEP LEARNING MODEL FOR UAVS"
_ICLR.cc/2019/Conference_

### Official Review · AnonReviewer1 · 2018-11-05
**Unclear contribution (did not implement in hardware, cited paper already did similar comparison of architectures)**

**Rating:** 2
**Confidence:** 2

**Review:**

The main context for this paper is two recent publications: Giusti et al.’s "A Machine Learning Approach to Visual Perception of Forest Trails for Mobile Robots” (2016) and Smolyanskiy et al.’s "Toward Low-Flying Autonomous MAV Trail Navigation using Deep Neural Networks for Environmental Awareness” (2017).

Giusti introduced a dataset of trail images (later called the “IDSIA dataset”) acquired by having a hiker wear three head-mounted cameras. The forward facing image is associated with a label “go straight”, whereas the two side images are associated with labels for “go left” and “go right”. Giusti then trained a convolutional neural network to predict these labels and used the network to guide a "quadrotor micro aerial vehicle”.

Smolyanskiy improves on Giusti’s work by (1) gathering additional trail image data using three cameras mounted to face forward but with lateral offsets and (2) using this additional data to train a 6 output neural network (“Trailnet”) which predicts both view orientation and lateral offset. In addition, they also combined predicted pose relative to the trail with predictions of localized objects and a depth map for potential obstacles. They compared several neural network architectures for predicting the view angle on the IDSIA data as well as the closed-loop performance of each network in avoiding collisions while operating within a UAV on a previously unseen trail. Though Trailnet did not achieve the highest accuracy (84% vs. the max 92% achieved by ResNet-18), it was the only network that achieved 100% collision avoidance on their UAV test course.

This paper, "A CASE STUDY ON OPTIMAL DEEP LEARNING MODEL FOR UAVS”, attempts to evaluate two potentially better convolutional neural networks for UAV trail guidance. They fine tune pertained Inception-Resnet and MobileNet models to predict the IDSIA dataset. These then both achieve better accuracy on the IDSIA test set and were analyzed for inference time and power consumption. These two models are then run through a single simulated path, where both seem to perform adequately across 2 turns in the path.

This paper has a variety of essential flaws.

1. A large portion of the text is devoted to their hardware and UAV control but they were not able to actually run models on a physical UAV "due to a hardware bug we were facing with the FCU”.
2. The paper claims to "introduce to the best of our knowledge, a very first comparative study of three algorithms in order to find a better motion control of a drone for detecting a trail”. This is a confusing claim since a comparison of neural network architectures is a central part of the evaluation in the Smolyanskiy paper.
3. The higher accuracy on view orientation does not seem relevant since it was also achieved by Smolyanskiy et al. with networks that they then showed performed worse when combined with object detection, obstacle depth inference and combined controller.
4. The sentence "An important goal of our method is to demonstrate the effectiveness of low cost systems for the complex task of flying an autonomous drone” appears to have been plagiarized from “Learning to Fly by Crashing” (2017) which contains "A important goal of our method is to demonstrate the effectiveness of low cost systems for the complex task of flying in an indoor environment”.

---

### Official Review · AnonReviewer2 · 2018-11-11
**Interesting comparison between different SOTA CNN for UAV trail guidance, but seems weak in clarifying novelty.**

**Rating:** 3
**Confidence:** 2

**Review:**

The paper initiates a comparison between different SOTA convolutional neural networks for UAV trail guidance with the goal of finding a better motion control for drones. They use a simulator (but not a physical UAV)  to perform their experiments, which consisted on evaluating tuned versions of Inception-Resnet and MobileNet models using the IDSIA dataset, achieving good results in the path generated.

I think that the authors have perform an interesting evaluation framework, although not novel enough according to the literature. It is also great that the authors have included an explicit enumeration of all the dimensions relevant for their analysis (which are sometimes neglected), namely, computational cost, power consumption, inference time and robustness, apart from accuracy.

However, I think the paper is not very well polished: there are quite a lot of grammatical, typing and aesthetic errors. Furthermore, the analysis performed is an A+B approach from previous works (Giusti et al.2016, and Smolyanskiy et al, 2017) and, thus, it is hard to find the novelty here, since similar comparisons have been already performed. Therefore, the paper needs major improvements in terms of clarity regarding the motivations in the introduction.

Also, one third of the paper is devoted to the software and hardware architecture used in the study, which I think it would be better fitted in an appendix section as it is of no added scientific value. Another weakpoint is that the authors were unable to run their DNN models on a physical drone in real time due to a hardware bug... I think the paper would benefit from a more robust (real) experimentation since, as they are, the presented results and experiments are far from conclusive.

---

### Official Review · AnonReviewer5 · 2018-11-12
**No substantial contribution, and large portions of the paper are unnecessary**

**Rating:** 3
**Confidence:** 3

**Review:**

Summary:

This paper considers the task of trail navigation task recently explored by Giusti et al. and Smolyanskiy et al. The authors describe their setup for physical experiments with a drone, and compare three neural network architectures for trail navigation on the IDSIA dataset. Experiments in a simulator are also reported.


Good aspects of the paper:

The pairing of simulation with trail navigation is an interesting idea, though it is not explored much in this paper.


Bad aspects of the paper:

Although the presence of physical experiments is suggested by pages 3 and 4, there are no physical experiments actually reported in the paper. In Section 5, this is revealed to be due to a hardware bug. The authors should not include these descriptions if they are not tied to reported experiments.

One of the main contributions of the paper is stated to be the comparison between neural network architectures. The two architectures compared to the TrailNet model from Smolyanskiy et al. are selected for their performance on the ImageNet classification task, and are shown to outperform TrailNet on salient metrics. However, comparing only three architectures is a very small comparison, and is not much of a contribution to the research problem.

This paper does not introduce new methods for approaching the problem of trail navigation. In its current form, it is a small comparison of existing classification architectures on the IDSIA dataset.

The paper also contains a number of minor errors. For instance, in Table 2 there is a footnote that leads nowhere, “introduced in Sif” is cited incorrectly, “in recent times jet (2014)” is cited incorrectly, and the figures are grainy (this isn’t really an error, but do try to make figures crisp in the future, e.g. with pdf images).

---

### Meta-Review · Area_Chair1 · 2018-12-16
**not a well polished paper**

**Confidence:** 4
**Recommendation:** Reject

**Metareview:**

The paper compared between different CNNs for UAV trail guidance. The reviewers arrived at a consensus on rejection due to lack of new ideas, and the paper is not well polished.